# Metal Exposure, Smoking, and the Risk of COPD: A Nested Case–Control Study in a Chinese Occupational Population

**DOI:** 10.3390/ijerph191710896

**Published:** 2022-09-01

**Authors:** Li Ma, Xinxin Huo, Aimin Yang, Shuxia Yu, Hongxia Ke, Mingxia Zhang, Yana Bai

**Affiliations:** 1School of Public Health, Lanzhou University, Lanzhou 730000, China; 2Department of Medicine and Therapeutics, The Chinese University of Hong Kong, Prince of Wales Hospital, Hong Kong SAR, China

**Keywords:** association, metals, smoking, COPD, interaction

## Abstract

Chronic obstructive pulmonary disease (COPD) was the third leading cause of death worldwide in 2019, with a significant disease burden. We conducted a nested case–control study using data from the China Metal-Exposed Workers Cohort Study (Jinchang Cohort) and assessed the associations of exposure to metals and tobacco smoking with the risk of COPD. We used the logistic regression model and the interaction multiplication model to assess the independent and combined effects of heavy metal and smoke exposure on COPD. The cumulative incidence of COPD was 1.04% in 21,560 participants during a median of two years of follow-up. The risk of COPD was significantly elevated with an increase in the amount of tobacco smoked daily (*p* < 0.05), the number of years of smoking (*p*_trend_ < 0.05), and the number of packs of cigarettes smoked per year (*p*_trend_ < 0.01). Compared with the low metal exposure group, the adjusted OR was 1.22 (95% CI: 0.85–1.76) in the medium exposure group (mining/production workers) and 1.50 (95% CI: 1.03–2.18) in the high exposure group; smoking and metal exposure had a combined effect on the incidence of COPD (*p*_interaction_ < 0.01), with an OR of 4.60 for those with >40 pack-years of smoking who also had the highest metal exposures. Both exposures to metals and smoking were associated with the risk of COPD, and there was an interaction between the two exposures for the risk of COPD.

## 1. Introduction

Exposure to metal and metal compounds has substantial effects on human health. Some heavy metals, such as arsenic (As), cadmium (Cd), nickel (Ni), mercury (Hg), chromium (Cr), zinc (Zn), and lead (Pb) are major health concerns for the public [1]. Some metals, such as As, Cd, Cr, and Ni have been confirmed as group 1 carcinogenic to humans by the International Agency for Research on Cancer (IARC) [2]. Exposure to heavy metals may have adverse health effects, such as cardiometabolic diseases [3,4,5,6,7,8], neurotoxicity [9,10], and respiratory diseases [11,12,13]. Recent studies have implicated exposure to metals as being associated with a decline in lung function [14,15,16,17]. We previously observed higher mortality due to lung cancer, cor pulmonale, and silicosis in workers exposed to heavy metals versus the general population [18]. An increased risk of chronic obstructive pulmonary disease (COPD) in workers exposed to welding fumes was observed [19], in addition to a linkage between metal dust and COPD in a case–control study [20]. Epidemiologic studies implicated environmental tobacco smoke (ETS), secondhand smoke, and tobacco smoking as key risk factors for COPD [21,22,23]. Johannessen et al. found that children exposed to ETS had double the risk of developing COPD in adulthood than those not exposed to ETS [24]. The introduction of reduced-risk tobacco would substantially reduce COPD-related deaths [25]. However, prospective studies that consider the compound exposure of multiple metals on the risk of COPD in a large sample are lacking.

COPD, a worldwide public health challenge, is the third leading cause of death globally [26,27]. In China, the prevalence, mortality, and disease burden of COPD have continuously increased over the past decades. The prevalence was 13.7% from 2012 to 2015 among 50,991 adults aged 40 years or older in a national cross-sectional study [27,28].

In this study, we investigated whether exposure to metal and tobacco smoking was associated with an increased risk of COPD in the China Metal-Exposed Workers Cohort Study (Jinchang Cohort), a large prospective study of unique metal exposures in occupational workers [29].

## 2. Materials and Methods

### 2.1. Study Population

A nested case–control study was carried out using data from the Jinchang Cohort, an ongoing prospective cohort study with ~45,000 metal-exposed workers aged 20+ years in Jinchang Nonferrous Metals Corporation (JNMC). They are routinely exposed to heavy metals, including nickel, copper, and cobalt, and the chemical processing of materials, including dust and gaseous pollutants. The details of the design and methods have been described elsewhere [29]. Briefly, a total of 42,122 participants (61.7% men, and 38.3% female) were included in the cohort. The baseline survey was conducted between June 2011 and December 2013, and 33,355 participants completed the first follow-up between January 2014 and December 2015.

The cases included all participants newly diagnosed with COPD in the Jinchang cohort during the first follow-up period; all cases were self-reported, and the patients had to provide the date of diagnosis by clinicians at the Workers’ Hospital of Jinchuan Group Co., Ltd., Jinchang, China). Controls were selected from the Jinchang cohort who were free of COPD and other respiratory diseases during the same study period. We matched 4:1 controls and cases by age (±3 years) and sex. We excluded those newly enrolled workers during the follow-up in both the case and control groups. Figure 1 shows the process of participant selection in this nested-case control study. This study was approved by the Ethical Committee of the Public Health School of Lanzhou University, and all of the participants provided informed consent.

### 2.2. Data Collection

Several types of data were collected from all participants, including in-person interviews, comprehensive physical examinations, laboratory tests, and biosample collection (all participants provided blood samples, and 20% of participants provided urine samples). The personal interviews were conducted by trained interviewers through a standardized and structured questionnaire. The information included demographic, socioeconomic, lifestyle, dietary, working history, occupational exposure, self-reported medical history, family history of disease, stress, and psychological status. 

### 2.3. Covariates and Smoking and Metal Exposure Assessment

We assessed smoking history via personal interviews by trained interviewers using a standardized and structured questionnaire that included an evaluation of their current smoking status, the age they started smoking and the average number of cigarettes smoked per day. Smoking status was categorized as current, former, and nonsmoker. Current smoking was defined as having smoked more than 100 cigarettes before the time of the interview and currently smoking cigarettes. Former smoking was defined as those who smoked more than 100 cigarettes, but who had stopped smoking for at least the past six months. Nonsmoking was defined as the rest of the participants (if the participant recently started smoking, and had smoked less than 100 cigarettes, but currently smoking, he/she was regarded as a non-smoker) [30]. Smoking exposure was evaluated by pack-year (combining smoking duration and tobacco exposure), and one pack-year was defined as one pack or 20 cigarettes smoked every day over the course of one year [31]. Participants were categorized by the number of pack-years (≤20 pack-years, 21–39 pack-years, ≥40 pack-years). 

The Jinchang Industry contains nine kinds of production and auxiliary service units. The metal exposure level is different among the various occupational categories in each unit. The occupational health risk assessment method from the International Mining Association was combined with the sources of heavy metals, types of heavy metals, working time, the protection measures of the working environment and other labor hygiene data. Appendix A shows the assessment results of the health risk of heavy metal exposure in the Jinchang cohort. According to the unit’s classification and occupational categories, metal smelting workers have the highest health risk of heavy metal exposure, and the risk level is divided into extremely high (RR = 9.00). In addition, the health risks of heavy metal exposure of workers in concentrators, miners and technicians are extremely high. The health risk value for chemical raw material and product production workers is 270, which is divided into very high grades; beneficiation plant technicians, mining plant office service personnel, metal metallurgical environmental management workers, mechanical maintenance workers and transport engineering workers have a high health risk level; those engaged in service and light industry and transport engineering, metal rolling processing office service personnel have a low health risk. The situations of extremely high and very high were rarely found. All high, very high and extremely high were categorized as high groups. In brief, metal exposure levels were categorized as low, medium, and high groups. 

In order to validate the levels of relative metal exposure among different occupational groups, we measured the urinary levels of nickel (Ni), cobalt (Co), copper (Cu) and 15 other heavy metals by inductively coupled plasma mass spectrometry (iCAP Q ICP-MS, Thermo Scientific, Waltham, MA, USA) in a selected subgroup of 500 workers from the biobank of the Jinchang cohort. We used the standard Reference Material Human Urine (SRM2670a, National Institute of Standards and Technology, Gaithersburg, MD, USA) as an external quality control, and we used sample spike recoveries to confirm the analytical recovery, which was 95%. The subgroup of 500 participants was aged from 20 to 50 years, half men and half women, and was matched by age and sex among three groups based on their occupation. There were 100 office workers (including service staff and managers) with presumed low levels of metal exposure, 200 mining/production workers (including chemical and metal products manufacturing workers, and the workers involved in mining and ore dressing) with presumed medium levels of metal exposure, and 200 smelting/refining workers with presumed high levels of metal exposure. The same detailed method of occupational metal exposure assessment has been described elsewhere [32,33].

### 2.4. Statistical Analysis

We calculated the cumulative incidence, in which the numerator was the new cases of COPD during the first follow-up, and the denominator was the total participants without COPD during the same period. We used the Kruskal–Wallis test to compare the urine levels of metals between the three occupational categories. We used conditional/multivariable logistic regression analysis to calculate the odds ratios (ORs) and 95% confidence intervals (CIs) for the association of metal and smoking exposures with the risks of COPD, adjusting for potential confounders including age, sex, race, educational level, working years, marital status, and body mass index (BMI, normal weight [<25 kg/m^2^], overweight [25–29.9 kg/m^2^], and obese [≥30 kg/m^2^]). The interaction effects between metal exposure and tobacco smoking were evaluated by including cross-product terms in the model. All analyses were performed using SAS version 9.3 (SAS Institute, Cary, NC, USA). All reported p-values were based on two-sided tests with a significance level of 0.05. 

## 3. Results

The cumulative incidence of COPD was 1.04% (224/21,560) during the period of the first follow-up of two years among 21,560 participants without COPD history. A total of 1120 participants were included in this nested case–control study, including 224 cases and 896 controls. 

The characteristics of the COPD cases and matched controls nested within the Jinchang cohort are summarized in Table 1. There were 725 males (64.73%) and 395 females (35.27%) in this study, wherein 97.68% were Han Chinese, 31.61% were office workers, 39.37% were mining/production workers, and 29.02% were smelting/refining workers. The mean age was 57.43 ± 2.58 years. All characteristics except education level and occupation did not show significant differences between the cases and controls.

Table 2 shows the urinary nickel, cobalt, and copper concentration in the 500 workers who verified the definitions of low, medium, and high exposure to metal in the Jinchang Cohort. The median concentration of urinary nickel, cobalt, and copper was 3.26 ug/L Creatinine, 0.43 ug/L Creatinine, and 12.16 ug/L Creatinine in office workers, 4.03 ug/L Creatinine, 0.52 ug/L Creatinine, and 11.33 ug/L Creatinine in mining/production workers and 4.55 ug/L Creatinine, 0.49 ug/L Creatinine, and 11.30 ug/L Creatinine in smelting/refining workers, respectively. The levels of nickel were significantly different among the three occupational categories, whereas there was no significant difference in both urinary cobalt and urinary copper between the three occupational categories. However, the levels of nickel, copper, and cobalt were significantly different among the three occupational categories in the selected male workers (*p* for all < 0.01). The concentration details have been described elsewhere [31].

Table 3 shows the associations between smoking exposure and the odds of COPD. Fewer than half (*n* = 548) of the participants were classified as current or former smokers. The adjusted OR was 1.51 (95% CI: 1.11–2.05) for those who were current or former smokers compared to nonsmokers (*p* < 0.01). The risk of COPD was also significantly increased with the increase in the number of cigarettes smoked daily (*p* < 0.05) and years smoked (*p* for trend < 0.05). After adjusting for potential confounders, a significant positive trend (*p* for trend < 0.01) was observed between COPD and an increase in the pack-years of smokers versus nonsmokers: adjusted OR = 0.57 (95% CI: 0.34, 0.96) for the lowest exposure (pack years < 20); adjusted OR = 1.66 (95% CI: 1.11, 2.46) for the medium exposure (20 < pack years < 40); adjusted OR = 3.11 (95% CI: 2.04, 4.74) for the highest exposure (pack years > 40).

Table 4 shows the association between metal exposure and the odds of COPD by smoking status. The adjusted OR was 1.22 (95% CI: 0.85–1.76) in the medium group (mining/production workers) and 1.50 (95% CI: 1.03–2.18) in the high group (smelting/ refining workers), compared with the low group (office workers). However, statistically significant associations were not observed based on occupational category between both current/former smokers and nonsmokers (*p* > 0.05). 

Table 5 shows the multiplicative interaction between smoking and metal exposure for COPD within the Jinchang Cohort. A statistically significant joint effect between metal exposures and smoking was observed in the prevalence of COPD (*p* interaction < 0.01). The OR of high metal exposure was 1.62 in nonsmokers but 4.60 in the heaviest smokers. The highest prevalence of COPD was observed for the office workers who had the highest pack-years.

## 4. Discussion

In this study, we examined the interaction of exposures to heavy metal and tobacco smoking on the risk of COPD in occupational workers in a large prospective cohort in China with two years of follow-up. We observed that both occupational metal exposure and higher smoking exposure were individually associated with an increased risk of COPD. The highest risk of COPD was observed in those with the highest exposures to both metals and smoke. Meanwhile, the joint effects between smoking and metal exposure levels were observed for the risk of COPD. 

Tobacco smoking was first confirmed as an independent risk factor for COPD in 1955 by Oswald and Medvei [34]. Much research has found that exposure to tobacco smoke is significantly associated with an increased incidence of, mortality from, and prevalence of COPD. In addition, a review presented that smoking cessation could reduce the decline in lung function and stop the progression of COPD, as well as increase the survival rate and reduce morbidity [35]. The adverse effects of smoking for COPD observed in this study were in line with prior studies, a significant positive trend was observed between COPD and an increase in pack-years in this study.

Meanwhile, another study showed that more than 50% of the cases of COPD were attributed to nonsmoking risk factors and up to 30% of COPD cases were attributed to occupational exposure [34]. The current study indicated a similar association between metal exposure and the odds of COPD. With the increase in the level of metal exposure, the risk of COPD was elevated. Moreover, a cohort study explored the relationship between occupational exposure to biological dusts, mineral dusts, gases/fumes, either vapors, gases, dusts, or fumes (VGDF) and moderate severity of COPD [36]. Tynes et al. reported that the risk of respiratory symptoms from exposure to metal dust/fumes was OR = 1.9 (95% CI: 1.3–2.7), and the joint effect of smoking and metal dust/fumes was OR = 7.0 (95% CI: 4.0–12.1) [37]. Hu et al. indicated a significant dose-dependent reduction in lung function and increased risks of COPD in coke oven workers in southeast China [38]. Gogoi et al. demonstrated a significant elevation in the levels of plasma heavy metals in COPD patients compared to a control group [39]. A study on the relationship between occupational exposure and the clinical features of COPD showed that those with COPD reported more symptoms associated with occupational exposure, even if they had a history of smoking [40]. The European Human Biomonitoring Initiative (HBM4EU) review of the scope of COPD and environmental substances reported that exposure to Pb was associated with COPD, while Cd, Cr, and As may be associated with COPD and/or decreased lung function [41]. In this study, we did not determine which specific metals may lead to an increased risk of COPD. Therefore, the quantitative assessment of metal exposure and the relationship between individual metals and the risk of COPD will be further studied.

At present, patients with COPD are not exposed to one single substance in the environment, and it is particularly important to study the interaction between exposed substances [41]. Boggia et al. confirmed a significant interaction (*p* < 0.001) between smoking and occupational exposure on the risk of COPD and increased risk among workers exposed to both risk factors (ExpB:2.51) [42]. Blanc et al. found that smoking and occupation were powerful and interactive factors in developing COPD [43]. In addition, a study on the effects of thallium and smoking exposure on lung function suggested that smoking can enhance the harmful effects of high thallium on lung function, and the preliminary analysis was that this may be due to increased inflammation [44]. The results of this study were consistent with those of the above studies, indicating that smoking and heavy metal exposure had an interactive effect on the pathogenesis of COPD.

Our study had several strengths. Firstly, it was based on a large sample size prospective cohort study, and we achieved a whole assessment of lifetime occupations and smoking history. Secondly, we confirmed the association between metal exposure and COPD among people who have never smoked, thus ruling out any potential confounding effect of smoking exposure and COPD. These results suggested that studies among people who have never smoked should be performed when evaluating health outcomes in relation to metal exposure. Thirdly, we compared the characteristics between 224 cases of COPD and 896 controls in the nested case–control group and adjusted the potential confounders. 

Nevertheless, there were some limitations that need to be considered. Firstly, the diagnosis of COPD was based on self-report in this study, which will result in some potential bias for confirmation with new cases of COPD and the lack of data on forced expiratory volume in 1s (FEV1)/forced vital capacity (FVC) for all participants. The patients who self-reported as having COPD were asked to have the diagnosis confirmed at the Workers’ Hospital of Jinchuan Group Co., Ltd., and they provided the accurate time of diagnosis to minimize the diagnostic bias. Secondly, the individual metal exposure was qualitatively quantified due to the lack of data on metal levels in the workplace air and the urinary metal levels of 80% in participants. The urinary metal levels were determined in a small subsample of three occupational categories to reduce the exposure misclassification. Finally, the sample size of this study was relatively small. More meaningful research results will be further expected within this ongoing prospective study as follow-up accrues. Furthermore, heavy metals as one of the harmful chemicals in tobacco smoke may be accumulated in tissues and fluids after smoking. So more meticulous research is needed to confirm the dose–response effect.

## 5. Conclusions

In conclusion, in this nested case-control study using data from the Jinchang Cohort, associations between exposure to metals and tobacco smoking with the risk of COPD were found. Concerning the adverse effects due to the interaction between exposure to metals and smoking on the risk of COPD, future prospective studies are necessary to confirm the dose–response relationship between the urinary/blood concentrations of the specific metals in workers and the risk of COPD to clarify the role of metal exposure and smoking in the pathogenesis of COPD.

## Figures and Tables

**Figure 1 ijerph-19-10896-f001:**
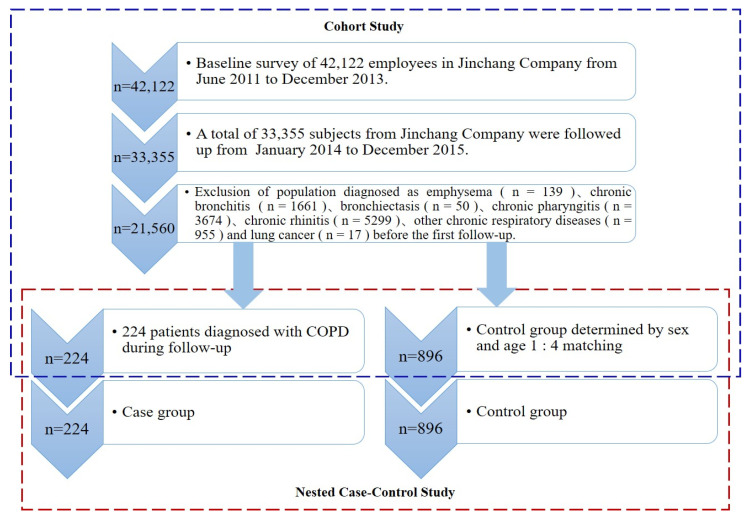
Flow chart of inclusion of the research population.

**Table 1 ijerph-19-10896-t001:** Distribution of the characteristics of the COPD cases and matched controls nested at first follow-up period within the Jinchang Cohort Study membership.

Characteristics	Cases (*n* = 224)	Controls (*n* = 896)	*p*
**Sex**			
Male	145	580	
Female	79	316	*p* > 0.05
**Age group**			
21–34	8	33	
35–49	60	205	*p* > 0.05
50–65	89	411	*p* > 0.05
65~	67	247	*p* > 0.05
**Race**			
Han Chinese	218	876	
Hui Chinese	5	12	*p* > 0.05
Other ethnic groups	1	8	*p* > 0.05
**Education level**			
No normal education	12	70	
Primary school	54	194	*p* > 0.05
Middle school	68	231	*p* > 0.05
High school	60	160	*p* < 0.05
College or higher	30	241	*p* > 0.05
**Working years**			
0~9.99	15	52	
10~19.99	19	68	*p* > 0.05
20~29.99	90	350	*p* > 0.05
30~	100	426	*p* > 0.05
**Marital status**			
Never married	3	18	
Married	192	779	*p* > 0.05
Remarried	3	16	*p* > 0.05
Divorced	6	25	*p* > 0.05
Widowed	20	58	*p* > 0.05
**BMI (kg/m^2^)**			
~25	137	576	
25~29.9	79	297	*p* > 0.05
29.9~	8	23	*p* > 0.05
**Occupational classification**			
Cadre	30	228	
Worker *	167	585	*p* < 0.01
Technical staff	7	27	*p* > 0.05
Office service staff	20	56	*p* < 0.01

* Workers are defined the mining/production workers (including chemical and metal products manufacturing workers and the workers involved in mining and ore dressing), the smelting workers and the refining workers.

**Table 2 ijerph-19-10896-t002:** The concentrations of major urinary metals among three occupational categories in the Jinchang cohort (ug/L Creatinine).

Occupational Categories	N	5th	25th	Median	75th	95th	H *	*p* *
**Nickel**								
Low	100	1.26	2.14	3.26	5.70	14.51	14.84	0.001
Medium	200	1.24	2.35	4.03	5.82	11.49		
High	200	1.30	2.73	4.55	8.28	30.79		
**Cobalt**								
Low	100	0.19	0.27	0.43	0.79	2.25	3.10	0.213
Medium	200	0.19	0.30	0.52	0.95	2.35		
High	200	0.19	0.31	0.49	1.01	2.61		
**Copper**								
Low	100	7.27	9.66	12.16	15.53	40.95	3.50	0.174
Medium	200	6.59	8.70	11.33	14.69	24.33		
High	200	6.26	8.75	11.30	15.18	28.66		

* Using the Kruskal–Wallis test.

**Table 3 ijerph-19-10896-t003:** Adjusted OR and 95% CIs for the associations between smoking and the prevalence of COPD within the Jinchang Cohort Study.

Smoking Category	Cases	Controls	Unadjusted OR (95% CI)	*p*	Adjusted OR (95% CI)	*p*
**Nonsmokers ***	94	478	1.0 (Reference)			
**Current/former smokers**	130	418	1.58 (1.18–2.13)	0.00	1.51 (1.11–2.05)	0.00
**Cigarettes smoked daily**						
~20	110	359	1.56 (1.15–2.12)	0.00	1.48 (1.08–2.03)	0.02
20~	20	59	1.72 (0.99–3.00)	0.054	1.68 (0.95–2.98)	0.08
*p* for trend			*p* < 0.01		*p* < 0.05	
**Years smoked**						
~15	13	36	1.84 (0.94–3.59)	0.08	1.81 (0.90–3.62)	0.10
16~30	35	139	1.28 (0.83–1.97)	0.26	1.28 (0.82–1.99)	0.28
30~	82	243	1.72 (1.23–2.40)	0.00	1.59 (1.12–2.26)	0.01
*p* for trend			*p* < 0.05		*p* < 0.05	
**Pack-years**						
~20	20	161	0.54 (0.31–0.94)	0.04	0.57 (0.34–0.96)	0.04
21–39	53	155	1.76 (1.21–2.55)	0.00	1.66 (1.11–2.46)	0.01
40~	57	102	2.84 (1.92–4.21)	0.00	3.11 (2.04–4.74)	0.00
*p* for trend			*p* < 0.01		*p* < 0.01	
**Age of starting smoking**						
~20	69	213	1.65 (1.16–2.34)	0.00	1.51 (1.05–2.17)	0.026
21–30	54	180	1.53 (1.05–2.22)	0.03	1.52 (1.03–2.24)	0.034
30~	7	25	1.42 (0.60–3.39)	0.42	1.35 (0.56–3.27)	0.51
*p* for trend			*p* < 0.05		*p* = 0.08	

* Referent group in all sections is ‘nonsmokers’.

**Table 4 ijerph-19-10896-t004:** Adjusted ORs and 95% CIs between heavy metals and prevalence of COPD within the Jinchang Cohort Study membership, stratified by smoking status.

Occupational Category	All				Current and Former Smokers				Nonsmokers			
	Cases	Controls	OR (95% CI)	*p*	Cases	Controls	OR (95% CI)	*p*	Cases	Controls	OR (95% CI)	*p*
Low	60	294	1.0 (Reference)		28	85	1.0 (Reference)		32	209	1.0 (Reference)	
Medium	88	353	1.22 (0.85–1.76)	0.28	56	205	0.83 (0.49–1.39)	0.48	32	148	1.41 (0.83–2.41)	0.21
High	76	249	1.50 (1.03–2.18)	0.04	46	128	1.09 (0.63–1.88)	0.78	30	121	1.62 (0.94–2.80)	0.09
*p* for trend		*p* > 0.05				*p* > 0.05			*p* > 0.05			

**Table 5 ijerph-19-10896-t005:** Multiplicative interaction between metal and smoking exposure for prevalence of COPD within the Jinchang Cohort Study membership.

Pack-Years	Occupational Category, OR (95% CI)					
	Low (Office Workers)		Medium (Production Workers)		High (Smelting/Refining Workers)	
	OR (95% CI)	*p*	OR (95% CI)	*p*	OR (95% CI)	*p*
**Nonsmokers**	1.0 (Reference)		1.41 (0.83–2.41)	0.17	1.62 (0.94–2.80)	0.12
~20	1.12 (0.44–2.87)	0.90	0.58 (0.23–1.44)	0.70	0.90 (0.39–2.06)	0.24
21–39	1.55 (0.68–3.50)	0.13	2.21 (1.23–3.97)	0.00	2.89 (1.50–5.56)	0.00
40~	2.08 (0.97–3.86)	0.00	2.59 (1.43–4.70)	0.00	4.60 (2.29–9.21)	0.00
*p* for trend	*p* < 0.01		*p* < 0.01		*p* < 0.01	
*p*-interaction	*p* < 0.01					

## Data Availability

The datasets used and/or analyzed during the current study are available from the corresponding author on reasonable request.

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
