# Peer review of "Metal Exposure, Smoking, and the Risk of COPD: A Nested Case–Control Study in a Chinese Occupational Population"

_ijerph, 2022, doi:10.3390/ijerph191710896_

Round 1

Reviewer 1 Report

The authors assessed metal exposures, smoking and risk of COPD in a Chinese occupational population.

Below are my queries and comments which needs to be addressed by the authors.

1.       Many issues around a lack of clarity in the manuscript result from imprecise wording and punctuation. English language editing and spell checking is required.

2.       Methods

More information needs to be included on how smoking was defined. For example, current smoking was defined as having smoked more than 100 cigarettes until the time of interview. If my interpretation of this is correct, the authors need to indicate that they are referring to the time the individuals started smoking until the time of the interview, and currently smoking.

4.       Current smoking was defined as having smoked more than 100 cigarettes until the time of interview and currently smoking cigarettes- has this method been verified in the literature. How did the authors come up with the cut-point of 100 cigarettes. This must be described in the methods.  Also does this mean If you recently started smoking, and had smoked less than 100 cigarettes, but currently smoking- you were regarded as a non-smoker? More details are required here.

5.       What about passive smoking, was this not taken into consideration?

6.       Table 1: occupational classification ‘worker’ needs to be defined in the foot notes.

Reviewer 2 Report

Comments to authors:

The manuscript entitled “Metal exposures, smoking and risk of COPD…” by Ma et, al. identified associations of metal exposure and smoking to COPD. The design, analyses, and presentation are logic, clear, and easy to follow. There is just one major concern to address, before this manuscript gets suitable to be published.

Major comment: metal exposure and smoking are not independent factors. Several heavy metals found in tobacco smoke, such as Cd, Cr, Co, Pb, and Ni. There are reports/studies showing that heavy metals, listed above, accumulate in tissues and fluids after smoking. Basically, the interaction between two exposures on risk of COPD may come from metals in smoking, as a result of dose effect. The design/analyses/interpretation of the results should take this factor into serious consideration.

Reviewer 3 Report

The article is very interesting and relevant.

When reading it, one is bound to have some doubts, which are related to the absence of some more detailed explanations, referring them to references to other articles.

Furthermore, it is unclear if all the people's occupational and personal factors have been considered.

Some suggestions for improvement can be found in the attached file.

Reviewer 4 Report

The authors conducted a nested case-control study using data from the China Metal-Exposed Workers Cohort Study (Jinchang Cohort) and assessed the associations of exposure to metals and tobacco smoking with the risk of COPD. Logistic regression model and the interaction multiplication model to assess the independent and combined effects of heavy metal and smoke exposure on COPD are studied.

A very interesting study, presenting a significant dataset, being studied a cohort and association of different variables of interest. However,  the characterization of occupational exposure to heavy metals is completely absent. In my opinion, the authors must include data/characterization of the tasks mostly performed by the classes of workers studied, as well as data on the period of exposure (8h daily, continuous, or shift work, with longer periods, years of working in this task, exposure frequency, etc). In the absence of quantitative data on IAQ and on biomonitoring of workers, the study will need to be more consistent regarding the characterization of human exposure to heavy metals and the potential development of COPD.
